# Current Situation and Construction of Recycling System in China for Post-Consumer Textile Waste

**Binbin Xu** [1], **Qing Chen** [2,*], **Bailu Fu** [2], **Rong Zheng** [2] and **Jintu Fan** [3]

1   College of Fashion and Design, Donghua University, Shanghai 200051, China
2   Shanghai International Fashion Innovation Center, Donghua University, Shanghai 200051, China
3   School of Fashion and Textiles, The Hongkong Polytechnic University, Hong Kong, China
*   Correspondence: chenqing@dhu.edu.cn; Tel.: +86-18261589352

**Abstract:** Waste recycling is an effective way to improve waste management, which is closely related to the support of social and economic foundations. With the development of a circular economy, green consumption is imperative. Most of the environmental protection brand enterprises are now almost limited to the environmental protection of clothing raw materials. However, there are still many problems in the overall industrial chain of the clothing industry, such as the pollution in the processes of processing, transportation and laundry, and the waste of resources caused by a large amount of textile waste after consumption. Starting from the theme of environmental protection and sustainable development of the clothing industry, this paper discusses the necessity of building a recycling system for post-consumer textile waste. Through the investigation of the recycling and reuse of domestic post-consumer textile waste, the existing problems are analyzed, such as the recycling supervision mechanism's imperfections, the trust crisis and a lack of recycling channels. Combined with the successful cases abroad, some solutions and suggestions are put forward for the regeneration and reuse of post-consumer textile waste, and a preliminary conception of the charitable market system is made.

**Keywords:** post-consumer textile waste; recycling; environmental protection; online; charity market system

## 1. Introduction

The waste of resources and environmental pollution in the textile industry cannot be underestimated. According to the literature data, the textile industry accounts for 5% of the total global waste [1]. According to the 2018 United Nations Economic Commission for Europe (UNECE) sustainable fashion report, the fashion industry emits 20% of the global total waste and 10% of the total carbon dioxide released, exceeding all international flights and ocean shipping's sum of emissions [2]. In China, 35 million tons of textile fibers are consumed each year, and it is estimated that about 20 million tons of waste textiles are produced each year, of which chemical fibers account for about 70% [3].

In addition to the huge waste generated during the manufacturing process, the waste of second-hand clothing and textiles has become one of the most serious resource wastes in modern society. From the 1960s to 2018, the annual generation of post-consumer textile waste (PCTW) in the United States increased nearly tenfold [4]. Many manufacturers in global supply chains and networks are in developing economies [5]. As a major manufacturing country in the world, China is facing the dilemma of serious environmental pollution and massive waste of resources. More than 70% of the old clothes recovered by China's old clothes recycling companies are exported for resale, the proportion of environmental protection recycling accounts for only 20%, and the proportion of donations in mountainous areas accounts only for 10% [6]. In an interview with Guo Dingyuan, a reporter from China Economic Herald, Guan Aiguo, president of the China Renewable Resources Recycling

Association, said: "The recycling rate of discarded clothing is less than 10%. It can consume half of Daqing's oil within one day".

In addition, due to the increasing use of synthetic fabrics containing various fiber blends in textiles, the difficulty of recycling is greatly increased [7]. In China, landfill or incineration are generally the methods chosen to deal with these old clothes. In the incineration process, waste textiles release a large amount of toxic gases, which causes secondary environmental pollution [8,9]. The China National Textile and Apparel Council issued the "Guiding Opinions on the Green Development of the Textile Industry during the 14th Five-Year Plan", and also pointed out that since the "13th Five-Year Plan", the green development of the textile industry has made certain achievements, but there are also many problems in development and issues that require ongoing attention. These include the slow construction of the waste textile recycling industry chain system, etc. [10].

Making full use of the recycling resources of post-consumer textile waste and building a complete and sustainable green supply chain system will greatly reduce the waste of resources and environmental pollution. The recycling system established in this paper mainly targets post-consumer textile waste (PCTW), which is composed of textiles that are no longer wanted by the owner [11–13]. The key goal in realizing a circular economy is to reduce waste and recycle resources [14], which includes making full use of post-consumer textile waste (PCTW) and building a complete and sustainable green recycling system. Therefore, it is necessary to conduct in-depth research on how to construct a recycling and reuse system of post-consumer textile waste.

## 2. Domestic Status of Post-Consumer Textile Waste Recycling and Reuse

### 2.1. Status Quo of Recycling and Reusing Post-Consumer Textile Waste in China

After investigation, Liu Yong Mei and others learned that about 15% of the post-consumer textile waste recycled by Shanghai Yuanyuan Industrial Co., Ltd. (one of the environmental protection projects implemented by the Shanghai Municipal Government) meets the donation conditions and will be directly handed over to the relevant departments for donation. About 40% of the post-consumer textile waste with no use value needs to be re-fibered, and about 15% of the clothing is exported to Africa and other places after disinfection treatment. These data also represent the current situation of domestic post-consumer textile waste recycling and reuse to a certain extent [15].

Second, most households in China are concerned about the hygiene of second-hand clothes, thinking that it is difficult to remove bacteria from old clothes with general cleaning methods [6]. In addition, because it is impossible to properly deal with the hygienic problems of recycled clothing, the trading market of second-hand clothing in China is very small. In addition to putting post-consumer textile waste in the used clothing recycling bin, most households' post-consumer textile waste goes directly into the trash can. With the continuous development of the social economy, the living standards of the Chinese people have also continued to improve. Since the 18th National Congress of the Communist Party of China, China has launched an all-out battle against poverty. The number of poor people in rural areas has been greatly reduced, and the demand for second-hand clothing in mountainous areas has also decreased year by year. The strategy of poverty alleviation has changed from direct money and goods to high-quality assistance at a precise technical level. Therefore, the domestic demand for charitable donations of second-hand clothing has dropped rapidly, and new recycling methods are urgently needed.

For the post-consumer textile waste exported abroad, it is first collected by the domestic post-consumption textile waste recycling company, and then the waste clothing is classified, cleaned and disinfected. According to the recycling statistics of Xianyu in 2018, in addition to the clothes donated to underdeveloped regions such as Africa, 45% of the old clothes are sold through trade [16] to countries including the Philippines, Cambodia, Pakistan, etc. The rest of the waste that does not meet export requirements will be reprocessed into simple textiles, such as rags and mops. However, many export companies aim at a profit,

and in the process of recycling, will cause many environmental and sanitation problems, which is contrary to the original intention of recycling old clothes.

### 2.2. Difficulties in the Recycling and Reuse of Post-Consumer Textile Waste in China

2.2.1. Lack of Recycling Channels

The current main recycling channels for Chinese consumers include: (1) Community-based clothing collection programs by local government agencies. For example, recycling bins or fixed recycling points; (2) Non-fixed charitable donation activities of various themes by non-profit organizations. Most of the recycled clothes in this channel are donated to impoverished areas after disinfection; (3) Companies such as Uniqlo and H & M collect their own brand (or any brand) of post-consumer textile waste from consumers; (4) Some consumers sell unwanted clothes through second-hand trading channels, such as idle fish platforms [17].

At present, most of the known recycling channels have some drawbacks. In terms of offline recycling channels, people have no way of knowing whether post-consumer textiles placed in the used clothes recycling bins are properly disposed of. Moreover, due to the size limitation of the throwing port, it is not conducive to the donation of large clothes. Most of the old clothes recycling bins are placed next to the trash can, and the sanitation environment is worrying. In terms of online recycling channels, certain online operation skills are required, and the scope of applicable people is limited, among which young users account for the majority. Additionally, some recycling channels have a deceptive purpose; in the name of charity recycling, they actually sell the recycled clothes for profit. In addition, the used clothes donation activities organized by charities are limited because they cannot be carried out for a long time, and most of the occasional activities cannot achieve the systematic long-term used clothes recycling system [18].

There are also a small number of initial systematic recycling companies in China. For example, Xi'an Aidishou Renewable Resources Co., Ltd. is one of them. Aidishou has a relatively complete recycling mechanism, including professional recycling vehicles, staffing, and store brochures. Under the background of the country's increased investment in the field of renewable resources and the encouragement of "Internet+"'s intervention in the traditional recycling industry, Aidishou also follows the trend of "Internet + waste recycling" and created Aidishou recycling APP. Figure 1 shows the recycling process of Xi'an Aidishou Renewable Resources Co., Ltd. However, the system network only focuses on the recycling level, in which there are major defects, and it has not been able to build a complete recycling and reuse system.

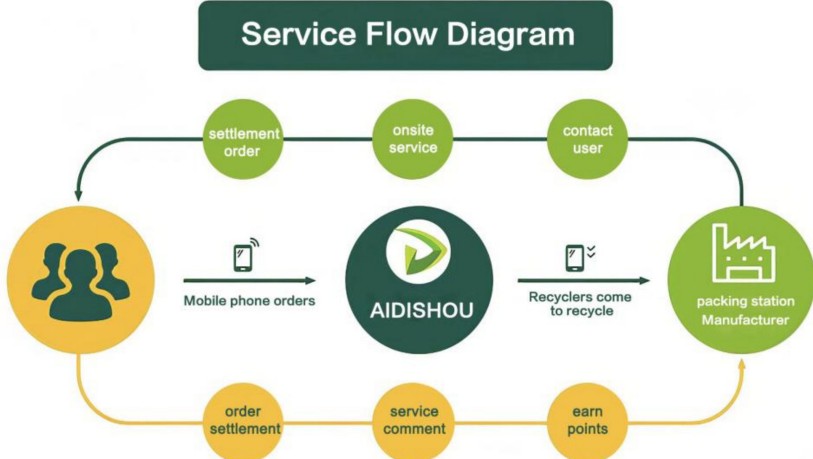

**Figure 1.** Recycling process of Xi'an Aidishou Renewable Resources Co., Ltd. Room 1602, 16th Floor, Unit 1, Building 12, Poly Zhongda Plaza, No. 303 Weiyang Road, Xi'an Economic and Technological Development Zone, Shaanxi Province.

### 2.2.2. Trust Crisis

People's trust in private recycling companies is low, and some large recycling companies will put the whereabouts of recycled old clothes on the platform to let the public know their location. However, some small-scale post-consumer textile waste recycling companies have a significant lack of systematic reports on whereabouts. Many companies will re-sell the old clothes donated by people under the guise of protecting the environment, thereby obtaining huge profits. In 2018, China became the fourth largest exporter of second-hand clothing. The price difference of this used clothing received at almost zero cost through reselling can directly skyrocket 20 times after arriving in Africa. Now many companies engaged in this industry can even achieve a daily turnover of 200,000. The news has taken a toll on consumers' trust in waste-clothing recycling companies.

### 2.2.3. The Recycling Supervision Mechanism

At present, China has established a complete recycling system and formulated corresponding policies for garbage disposal, but there is a blank in the recycling of old clothes. In the absence of supervision by relevant laws, regulations and mechanisms, the government cannot effectively supervise and manage them. As a result, the used clothes recycling market is relatively fragmented and chaotic. In order to gain vested interests, some companies sell old clothes and do false charity, which disrupts the market order.

### 2.2.4. Processing Methods

In terms of the recycling method of post-consumer textile waste, China is still relatively backward in the key technologies of recycling post-consumer textile waste. The mechanical recycling method is the most widely used treatment method in China. This recycling method does not need to separate the post-consumer textile waste, the operation is relatively simple, the pollution is small, and the cost is low, but the added value generated is relatively low.

## 3. Status Quo of Recycling and Reuse of Used Clothing Abroad

### 3.1. European Union

To improve the recycling participation of relevant enterprises, the EU requires product manufacturers to be responsible for the entire life cycle of the product, especially the recycling and disposal of the product after sale. After France adopted this policy for enterprises in 2007, about 35% of waste textiles were recycled, and 60% of the recycled textiles were reused. The enthusiasm, participation and responsibility of enterprises in recycling can be greatly improved through this policy, so that waste textiles can be better recycled and treated [19]. Therefore, many countries have implemented or are preparing to implement Extended Producer Responsibility (EPR), with Sweden launching it in January 2022 [20] and the UK government launching it in 2025 [21].

Whether recycled textiles are truly safe, environmental protection is also an important factor affecting consumers' acceptance of recycled products. The EU textile industry chain achieves this through the labeling system—the Nordic Swan Ecolabel. Consumers can scan the QR code on the product to know the source of the product, processing steps, color fastness and shape retention standards, water use, and information on emissions reductions to the air during production. This system can enhance the transparency and reliability of recycled product information, and improve the convenience, trust and motivation of consumers to purchase recycled products.

### 3.2. Switzerland

In Switzerland, there is a recycling company, Texaid, which was established in 1978. Its operation model is that the post office regularly mails special used clothes recycling bags to residents. Residents put old clothes in the recycling bag, and then place it outside their door. Company service personnel will collect them, or residents can drop them directly in a company-designated recycling station. These recycled clothes will be processed in

three ways, depending on the new, used and damaged level, namely direct sales, charitable donations or similar products, such as wipes.

### 3.3. Belgium

The Belgian company Earth has specially produced a recycling bag that is distributed to each family for free through the postal department every month. The collection time and items that can be stored, such as old clothes, shoes and hats, etc., are listed on the recycling bag. Residents can place their bag outside the door the night before the collection day, and the company will arrange for a special person to come and collect them.

Among the existing recycled old clothes in Belgium, the recycled clothes will be divided into three categories according to the degree of new, old and damaged: high-quality, reusable and unusable. A total of 60% of the old clothes will enter the factory for cleaning and disinfection, etc., and then be regenerated into coarse cloth or become a complete fiber. Knitted products made of these recycled fibers will appear in the European market in large quantities which realizes the reuse of post-consumer textile waste. The methods of Belgium and Switzerland are basically the same. After years of practice, they are systematic and worth learning.

### 3.4. Japan

Japan is limited by geographical environmental factors and resources are extremely scarce. Therefore, reducing resources and using them sustainably are the top priorities in Japan, which has prompted the nation to be concerned about environmental protection. Some Japanese ready-to-wear companies have gradually shown a thriving scene through the research on green and sustainable production.

H & M was the first fast fashion retailer to launch the Used Clothing Recycling Program (UCRP) in 2013, and the brand collected 20,649 tonnes of textiles for recovery and recycling in 2018 and 29,005 tonnes in 2019. UNIQLO's second-hand clothing recycling activities have also been warmly welcomed by consumers. From September 2019 to January 2021, the brand collected about 620,000 down jackets in Japan through recycling activities. From September 2020, the campaign will expand to 22 other markets overseas. Rapid expansion of used clothing recycling program proves growing customer demand for recycling services [22]. In today's era of increasing emphasis on environmental protection, UNIQLO's recycling of used clothing has brought a positive impact to the clothing industry. UNIQLO can not only attract more customers but also form a long-term potential customer base through its contribution to environmental protection and the recycling of specialty stores. At the same time, flexible recycling methods have also improved brand image.

## 4. Ways to Recycle and Reuse Used Clothing

### 4.1. Clothing Rental and Sharing Mode

Shared consumption, also known as cooperative consumption, means that consumers acquire the right to use and own goods through leasing, exchange, gifting, and other means, which can efficiently use resources and can generate strong economic mutual benefits [23,24]. The sharing model of the clothing and fashion industry mainly adopts the model of clothing leasing. China has also ushered in a boom in the online clothing rental industry since 2015, but with the neglect of consumers' real clothing needs, poor hygiene and cleaning conditions, lag in customer service communication services, and improper establishment of membership systems [25], the clothing rental platform's model faces increasing distrust and customer disputes. The post-2019 COVID-19 pandemic disrupted the global economy. According to WHO [26], COVID-19 is mainly transmitted through respiratory droplets, and the use of incompletely disinfected personal items increases the risk of contracting the virus. Although severely affected by the epidemic, numerous studies still maintain a positive attitude toward the clothing leasing industry. According to relevant studies, the leasing model is more sustainable than the traditional consumption model [27–29], as it can effectively reduce the excessive consumption of resources and

reduce costs, can use better preservation clothing, etc.. The U.S. company Rents The Runway lost numerous subscribers in 2020, but in 2022, its subscriber numbers returned to pre-pandemic figures [30,31]. However, the situation in China is just the opposite. Chinese consumers are more concerned about privacy, and they are more concerned about the hygiene and safety of clothing during the epidemic. Additionally, due to the impact of the epidemic on economic income, the results of a survey of 500 Chinese consumers by the Cotton Company [32] showed that the overall clothing expenditure ratio dropped by 69% after the epidemic. Under such a combination of reasons, most of the sharing platforms in China have a short lifespan or poor operation. Chinese local clothing rental platforms such as Goddess Pie, Clothes and YCLOSET have ceased operations, and Le Tote, an American clothing rental company, has also filed for bankruptcy protection in 2020. At least until the drastic impact of the epidemic disappears, the difficulty of operating clothing rental platforms in China will be very difficult (Table 1).

**Table 1.** Second-hand clothing rental platform survey and analysis table.

| Clothing Rental Platform | MSPARIS (Closed) | YCLOSET (Closed) | YIKU (Closed) | Le Tote (File for Bankruptcy Protection) |
|---|---|---|---|---|
| clothing variety provided | The category is relatively simple. | More big-name clothing, the main high-end clothing. | Fashion brand, small and fresh clothes, gowns and so on. Mainly for students. | The variety is relatively rich, the early big brand more, the late quality is general. |
| Membership system | Ordinary members 15 RMB/18 days, 8 RMB/7 days (only 4 clothes can be rented once) VIP members 120 yuan/18 days | The 499 RGB monthly pass allows you to rent three items at a time and unlimited times each month | The annual membership card is 2988 RGB | 299 RGB/month/6 pieces, 499 RGB/month/12 pieces, 599 RGB/month/12 pieces four accessories. |
| Garment price | 1500 RGB or so | 500 RGB or so | 200–500 RGB | 300–1000 RGB |
| the related evaluation | The category is relatively simple, the classification is not obvious, the customer service attitude is poor, the clothing hygiene condition is poor. | The category is relatively simple, the classification is not obvious, the customer service attitude is poor, the clothing hygiene condition is poor. | The membership fee is the cheapest, the customer service attitude is good, but there is deniability problem, the clothing is more daily, less quality problems, the new fast (no sign for 40 days after the return of clothes). | The clothing cleaning degree is worrying, the response and processing time of customer service is long, (management negligence, customer slander), the cost is expensive, and the membership time is more than 20 days before the rental of clothing. |
| conclusion | Quality clothes need to be snapped up; Uneven sanitary conditions; Delivery is too cumbersome and the logistics process takes too much time; China's user base is small; Improper establishment of membership fee system; Most of them have the problem that the clothes are not authentic. | | | |

### 4.2. Raw Material Recovery and Reuse Mode

In addition to the model of a clothing rental platform, recycling post-consumer textile waste to reduce waste of resources is the most widely used model at present, including energy recovery methods, mechanical recycling methods, physical recycling methods and chemical recycling methods. Among them, the mechanical recovery is the method with the widest application and the simplest operation, and the chemical recovery is the method with the most complete recovery and the least pollution (Tables 2 and 3).

**Table 2.** Introduction and analysis of recycling methods.

| Recycling Method | Concept | Advantages | Disadvantages |
|---|---|---|---|
| Energy recovery act | Usually, synthetic fibers with no utilization value but high heat content are selected, and consumer textile waste is used as fuel to convert heat energy into electrical energy for utilization [33]. | The recovery method is simple and low cost. | Efforts and funds paid for environmental protection recycling in the early stage are wasted, which will cause irreparable damage to the ecological environment [34]. |
| Mechanical recovery method | Mechanical recycling refers to the method of fabric recycling that does not sort the materials of post-consumer textile waste and does not destroy the structure of the fabric itself. There are two main directions. One is to make fibers with a certain use value without prior separation treatment, and then make them into yarn or textile fabrics. The products are mainly low-value products. The fibers made with this method can be used as filling materials, materials for heat and sound insulation layers, carpets, etc. The second is to simply process the recycled waste clothes into second-hand products with use value [33]. | There is no need to separate and dispose of old clothes, and the operation is relatively simple [33]. It has high utilization rate for waste textiles, wide application scope and lower investment. | The industrial scale is small and lacks of standardized industry management, which makes it easy to cause waste of scrap materials and secondary pollution in the process of separation and training. The added value of recycled products manufactured by this method is also low [35]. |
| Physical recovery method | Physical recycling is usually used to recover purely spun natural or synthetic fibers. After the post-consumer textiles are screened, the long fibers in the discarded clothes are converted into short fibers by sorting, cutting, and crushing without adding chemicals, and then the processed short fibers are re-spun into yarns, so that old clothes obtain new utilization value [33]. | The traditional physical recycling method has high recycling efficiency and little impact on the environment. | The classification is difficult, the technical content is high, and the recovery process is complex, so it is not suitable for blending, which will easily lead to short fiber length and mechanical properties' decline after regeneration [33]. |
| Chemical recovery process | The chemical recovery and regeneration technology is mainly aimed at waste polyester textiles, not suitable for natural fibers. It can be recovered from the final product and regenerated to achieve real recycling. A special chemical method is used, such as alcoholysis technology, sub-Critical depolymerization technology or the use of catalysts, such as enzymes, to depolymerize the high molecular polymers in waste clothes, so as to achieve the separation of fiber types, and then repolymerize these monomers to produce new textile fibers, which re-enter the spinning process and enter the textile industry in the next life cycle [33]. | At present, it is the best regeneration method, which has the advantages of deep decomposition and sustainable development, especially suitable for some valuable polymers. | The technical requirement is high, the equipment is expensive, the process is complex, the recovery cost is high. At present, it is difficult to realize industrialization [33]. |
| Analyzed | The energy recovery method is the easiest to operate and the lowest cost, but the harmful gases produced by combustion will cause a certain degree of pollution to the atmosphere. The chemical recycling method has little impact on the pollution of the environment and the degree of regeneration is the most thorough, but it faces problems such as high cost and high process requirements, and is not suitable for natural fiber materials, making it difficult to expand the scope of use. On the whole, each method has its advantages and disadvantages. The application should be based on the level of economic development of the area used, the type of clothing collected, the degree of newness, and the regeneration target of post-consumer textile waste. | | |

**Table 3.** Comparative analysis of causes related to recycling methods.

| Recycling Method | Energy Recovery Act | Mechanical Recovery Method | Physical Recovery Method | Chemical Recovery Process |
|---|---|---|---|---|
| Case | The processing line developed by SREAD converts waste clothing, textiles, leather products, etc., into high calorific value SRF/RDF, which can be used as a fuel instead of coal, thereby transforming waste into energy [36]. | The recycled fiber comber developed by Xinshunxing Recycled Clothing Technology Co., LTD converts waste textiles into renewable fibers after a series of processes, which can be made into rags, blankets and heat insulation MATS. | At H & M's mall store in Stockholm, the clothing-to-clothing recycling machine is a scaled-down version of an industrial model that can take apart, wash and chop old clothing into fibers that can be spun into yarn and woven into sweaters, scarves or baby blankets. | The cotton fibers in waste textiles can be directly depolymerized to produce glucose solution through acid hydrolysis, and then more than 90% glucose yield can be achieved through the two-step method of combining concentrated and dilute sulfuric acid [37]. |
| Analyzed | According to the cases cited, the application of energy recovery method, mechanical recovery method and physical recovery method in the production of recycled and reused products is relatively mature, while the chemical recovery method has many limitations due to its short development time, etc. The reason is that it is mostly a small-scale application in the laboratory. However, considering the added value of recycled products and the impact on the environment, the chemical recycling method still has great development prospects. | | | |

## 5. Suggestions on the Construction of Post-Consumer Textile Waste Recycling System

*5.1. Preliminary Infrastructure Construction of the Waste Textile and Clothing Textile Recycling System*

5.1.1. To Complete the Legal System Related to the Recycling of Post-Consumer Textile Waste

Waste management should play a central role in the government's environmental policy and a sound legal framework will also contribute positively to the development of waste management systems [38]. China needs to formulate regulations and mandatory regulations on the recycling of post-consumer textile waste. The first is to introduce certain preferential policies to encourage enterprises. Then, the propaganda of relevant legal concepts is strengthened. In addition, it is also essential to strengthen supervision. According to the survey, when the carbon allowance to reduce emissions is larger than the threshold, manufacturers will increase investment in green technologies while deciding on recycling [39]. Hence, the government plays a leading role in the upper-level construction of a post-consumer textile waste recycling market.

5.1.2. To Improve Public Trust and Attention

In the post-consumer textile waste recycling industry, there are some opportunistic companies, which have caused the public's lack of trust in post-consumer textile waste recycling companies. Therefore, it is very important to select companies with compliance management, devoted research, and continuous innovation. The government can strengthen cooperation with these companies by providing some preferential policies. Part of the income obtained by recycling companies can be used for public welfare and scientific research. Companies can also work with a reliable charity to build a long-term relationship to enhance the company's reputation. Tax breaks and incentives can effectively encourage companies to produce environmentally friendly materials. As a result, the problem of the recycling and reuse of post-consumer textile waste can be better solved through cooperation between the government and enterprises.

In addition to trust, the survey results show that the main determinant factor on consumers' willingness to participate in used clothing recycling programs is their attitude towards the environment. Consumers who care about the environment are more willing to participate [22]. Therefore, the promotion of government–enterprise cooperation can more effectively attract consumer's attention to the environment, thereby increasing their interests in recycling waste clothing.

### 5.1.3. To Establish a "Waste-Resource" Circular Chain

The traditional linear economy adopts a one-way structure of "resource-based products-wastes". H & M is an advocate of the concept of sustainable human development. Its closed-loop model under the circular economy increases the reverse logistics process of "goods–resources". From the selection of raw materials, green design, clean production, green packaging, and transportation, and by advocating green consumption and guiding customers to participate in recycling activities of second-hand clothing, the recycled clothing can be reused as raw material, and repeat the cycle on and on [40]. This circular chain also has great reference for the establishment of the recycling system of post-consumer textile waste with the charity market in this research. Figure 2 shows the H & M closed-loop mode.

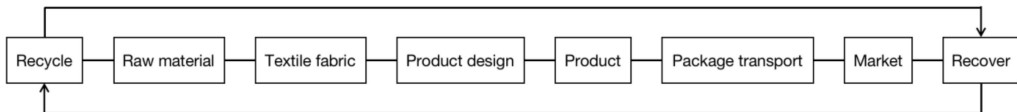

**Figure 2.** H & M's closed-loop mode.

### 5.2. Construction of Post-Consumer Textile Waste and Textile Recycling System with Charity Market

The government's support and subsidies for recycling institutions will also affect the quality of recycled waste products to a certain extent. When the government subsidizes processors, the commissioned recycling price of waste products will increase, but when the government subsidizes recyclers, it will decrease [41]. The policy of fund rescue can motivate producers' sustained innovation behavior [42]. Based on the investigation of some excellent cases of post-consumer textile waste recycling abroad, it can be found that governments of many countries have established professional charitable waste textile recycling organizations, which have played an important role in promoting the textile recycle circulation. By conducting a relevant questionnaire survey on Tianjin residents, it was found that 60% of the respondents were likely to purchase recycled clothing. However, 46.7% of the respondents have concerns about the hygiene and safety of recycled clothing. Therefore, enhancing the consumers' understanding and trust in the recycling process is the key to enlarging residents' purchase. In addition, 80.7% of the respondents have a strong interest in the intelligent service platform for household clothing recycling of old clothing. With the help of the state's encouragement of "Internet+" to intervene in the traditional recycling industry, the establishment of an Internet-based recycling information platform can not only meet the needs of residents, but also better fit national policies and improve the development of post-consumer textile waste textile recycling system [43].

Therefore, according to the current situation of China's textile industry, the establishment of a professional charitable organization under the supervision of the government, as a link between the public, charities, and recycling companies, will gradually form a complete industrial chain, which will better promote the ability of recycling processs in China [44]. Figure 3 shows the textile recycling system established by this study.

### 5.2.1. Recycling Link

Research indicates that under the influence of circular economy theory, a waste clothing recycling channel (WCRC) is an important link affecting the efficiency of waste clothing recycling. Therefore, it is very important to establish a more complete and efficient recycling channel [45].

In terms of recycling methods, it is concluded that supporting policies are more influential than economic factors in motivating consumers to recycle clothing [46]. Therefore, it is necessary to consider how to better aid consumers to involve the recycling process more conveniently. For example, special used clothes recycling bags with brochures can be distributed to each household through the China Post system. Due to the high reputation

of China Post, people's trust and participation can be enhanced with this recycling method. Consequently, residents can easily to participate in the recycling of process. Although clothing recycling bins are placed in the community, more effective process instructions and attractive visual cues are required outside bins. In addition, icons on bins are more effective and significant compared to posters and flyers [47]. Moreover, colleges and universities in neighboring areas cannot be ignored. Due to frequently discarded clothing by students, it is recommended to cooperate with the Youth League Committee and Student Union of each school for regular recycling.

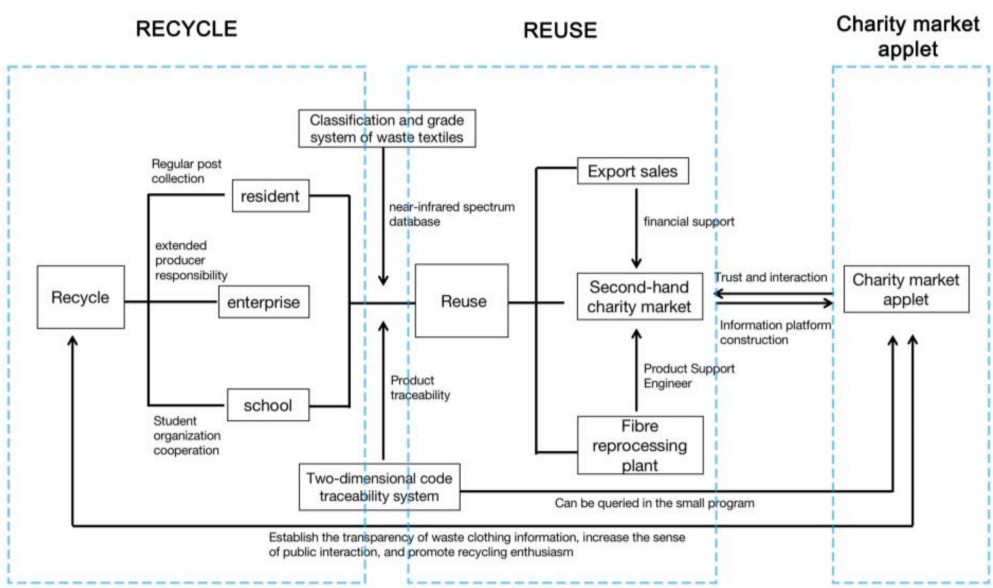

**Figure 3.** Textile recycling system.

In addition, awards can be presented to participants in this activity to encourage people to participate in this type of social initiative. Door-to-door recycling services should be arranged to reduce trouble for students and residents.

5.2.2. Reuse Link

In the treatment of recycled old clothes, minimizing production costs enables enterprises to maintain competition in the used clothing industry. On the other hand, maximizing recycling rates can reduce the environmental impact of post-consumer textile waste, which is an essential requirement. The first step is to systematically classify the old clothes from the collection process. The production cost and new material cost are estimated according to the type, wear condition (high, medium, low) and material (cotton, linen, wool, silk, rayon). The appropriate classification of the wear condition and application according to different reuse directions can generate the minimum cost and the maximum recycling profit and utilization rate [48]. It is important to establish the waste grade standard for textiles on the basis of the national waste recycling classification standard for all categories. In the follow-up, the near-infrared spectral library is used to classify post-consumer textile waste and textiles according to the classification level standard, so as to improve the efficiency of classification work and the level of automation and informatization. In addition, a domestic recycling QR code traceability system can be established to achieve the transparency and reliability of recycled products, provide consumers with relevant product traceability information, and increase consumers' trust and acceptance of second-hand recycled products through this system.

After the post-consumer textile waste is classified and marked, their processing directions can be roughly divided into three categories. According to the current data in China, the clothes with high quality in the recycled post-consumer textile waste will be donated to less developed areas, and the other clothes will be exported for resale to undeveloped

oversea areas and recycling. According to these classifications, there are several suggestions for the flow direction of post-consumer textile waste.

Domestic Second-Hand Market

In terms of impact-oriented behavior, avoiding over-consumption is one of the solutions to the problem of clothing sustainability. The purchase of second-hand clothing or rental of clothing are important solutions to the unsustainable clothing consumption. Although these behaviors are not generated from the consideration of sustainability, it could obviously avoid wasting resources and effectively reduce the excessive consumption of new clothing products [49]. As early as the 1980s, many second-hand clothing markets appeared in foreign countries, and some families sold their idle used clothes in the second-hand clothing market [50]. There are very few second-hand markets in China, mostly small flea markets or online platforms. The used clothes recycling trade has developed for the past 30 years. In the 1980s, the used goods in Beijing were relatively completely from acquisition to market sales. Over the past decade or so, Beijing has abolished the acquisition and sales system of used clothes to prevent the spread of infectious diseases [6]. The "Research Report on Recycling and Utilization of Waste Textiles in China" issued by China Textile Federation pointed out that one of the main problems of recycling and utilization of waste textiles in China at present is the imperfection of market promotion and the construction of a second-hand trading market. In addition, it pointed out that there are significant differences in consumers' perception of second-hand fashion products and services. Consumers do not accept second-hand clothing because of poor product quality and cleanliness [51]. China should formulate relevant regulations and standards to guide and standardize the recycling and reuse of waste textiles. For example, if the disinfection methods and standards of waste textiles are clearly defined, and the hygiene and quality of waste textiles are standardized, second-hand clothing stores will have a large market space.

Therefore, considering the current acceptance of second-hand clothing delivered to less-developed areas and incomplete second-hand market in China, it is recommended to carry out second-hand sales activities in the charity supermarket after cleaning collected old clothes. Due to the economy recession under the influence of the COVID-19 epidemic, second-hand clothing can be an attractive retail option for those with severe financial crisis [52]. Moreover, the charity market can also carry out cooperative design activities with brand enterprises to redesign and innovate this second-hand clothing. Gen Y consumers also hope that the fashion industry will actively engage in ethical trade by continuously improving production processes and supply chains [53]. Research shows that the uniqueness, high quality and fashion trends of used clothing are factors that influence whether clothing consumers buy used clothing [54], but in the process, brands need to consider how to ethically reuse or responsibly recycle the still usable textiles, rather than causing more waste [55]. It is also possible to increase the publicity of the event, attract more people to participate in the action of buying second-hand clothes, and add more value to the reuse of these second-hand clothes.

Second-Hand Clothing Export Industry Chain

Second-hand clothes exported from China to Africa and Southeast Asia have become a complete industrial chain. According to Yu Min Xing, a reporter from Qilu Evening News, the company (Qingdao Recycling Environmental Protection Technology Co., Ltd.) purchases post-consumer textile waste at a price of 1500 RMB per ton. After selection and pre-treatment, they are sold to Guangzhou companies at 5000 RMB per ton, and to Africa, Cambodia, East Timor, and other places at a high price of 7000–8000 RMB per ton. This profit from this business is higher than that from the sale in the domestic second-hand market and fiber remanufacture.

However, because these enterprises chase high profit, they can cause various environmental pollution and sanitation problems while recycling, which deviate from the original intention of recycling post-consumer textile waste. Therefore, these private enterprises

should become part of the charitable market under the supervision of the government to conduct business activities in order to reduce harm to the environment. A portion of the collected profits can also be used to carry out more charitable activities and support the operation of the charitable market.

Fiber Remanufacturing Plant

At present, China's post-consumer textile waste recycling technology is not perfect. Mechanical recycling is widely applied, but physical recycling and chemical recycling have relatively high utilization ratio and low pollution. The subcritical depolymerization technology in chemical recycling can restore waste textiles to the raw material level, and the regenerated products are of high quality and environmentally friendly [56]. In addition to sub-prosecution technology, ionic liquid dissolving agent is also an excellent catalyst for dissolving cotton textiles at present, which exhibits the advantages of easy separation and recovery, and high catalytic activity [57].

The government should encourage the research on key technologies for recycling post-consumer textile waste. In addition, policies such as technology promotion, innovation incentives, and tax relief can also be implemented to promote enterprises' participation in technological upgrading and innovation. It can also conduct cooperative research with companies that already have mature recycling technology.

After effective treatment of post-consumer textile waste textiles, these fibers can be used to make a variety of products. The fibers are consolidated into nonwovens by means of a needling technique without any adhesive. This material can be used in disposable products such as sanitary napkins, napkins and diapers [58]. Since its skeleton consists of fibers and pores, it is well suited for insulating applications [59–61]. Ground fibers can also be used for strengthening and viscosity modification of different materials, and cotton mixed with asphalt can improve the quality of asphalt [62]. This material can also be used as a durable material for buildings or automobiles for thermal and acoustic protection. According to the literature data, the production of nonwovens from textile waste requires fewer steps than yarn production, thus reducing the cost of synthesizing new materials and reducing the environmental impact [60]. Other textile products, such as blankets and scarves, can be sold directly at charity markets. It can not only reduce the resource usage, but also activate and manage the charity market.

## 5.3. Charity Market Mini Program

When it comes to recycling post-consumer textile waste, most people question whether the clothes donated are really properly used or whether the recycled products are safe products. With the negative news about philanthropy emerging in recent years, it is necessary to establish an information platform on how to solve the non-transparency and non-response during circulation process and increase the public's participation in post-consumer textile waste recycling activities.

According to a questionnaire survey, most residents believe that a second-hand clothing recycling public welfare project carried out in the form of the Internet can promote the development of used clothing recycling [63]. Related research on how to increase participants' willingness to recycle has found that information feedback has a more positive impact on activity participants' responses [64]. After surveying 238 U.S. consumers, the results showed that feedback on the history of second-hand products can enhance consumers' trust in services, and greater trust can enhance consumers' positive attitudes toward second-hand clothing products, thereby changing their perception of the use of second-hand clothing products [65]. Additionally, recycling messages can be disseminated to the wider community through social media influence and NGO intervention [66]. Therefore, it is necessary to establish an online information publicity feedback platform.

With the rapid development of the Internet, the "Internet + recycling" model is generated and developed. The use of some online platforms is very mature, such as WeChat Mini Programs log in directly from WeChat. It has the advantages of no registration,

no installation, and instant use. It can be quickly shared with friends and has higher interactivity. The itinerary code and health code in the epidemic era are carried on the small program platform.

By scanning the QR code on the product, its basic information can be indicated through the code traceability system. If the donated clothing is relatively new and meets the conditions for selling second-hand clothing, the donor will get a QR code, through which they can learn about the sale and whereabouts of the donated clothing. The income from the sale of this garment will also be converted into donation points, which can be exchanged for some small gifts. In addition, as mentioned in the recycling link, the information of each donor will be counted, and special rewards will be given according to the number of donors. Meanwhile, charitable market institutions can also carry out charitable cooperation with clothing brands, and the brands will provide gifts or brand coupon points, etc., which will help increase the positive image of the brand and the public's goodwill towards it. Moreover, charitable market institutions can also cooperate with Alipay and Taobao on initiatives such as using the charity market applet points to interconnect or exchange Alipay Ant Forest points, and the points can be exchanged for Taobao coupon.

Studies have shown that consumers' willingness to donate for others' benefit in charitable appeals is greater than their willingness to donate for their own interests [67]. As a result, The Charity Mini Program will also publicize the whereabouts of its operating income, showing its donation or assistance results, shaping better environmental awareness and recycling intentions, which have positive implications for green buying behavior in emerging economies [68]. Through the advantages of the internet in information collection, data analysis, flow monitoring, real-time sharing, etc., a more transparent, convenient and efficient online or offline platform is built [33].

*5.4. SWOT Analysis of Recycling System in China for Post-Consumer Textile Wasted*

Through the SWOT analysis method, we analyzed the reality faced by the recycling system in China for post-consumer textile waste, which consists of recycling links, reuse channels and online small programs in the charity market. An objective and comprehensive analysis conclusion of the system was obtained (Table 4).

**Table 4.** SWOT analysis of recycling system in China for post-consumer textile wasted.

| Strength | Weakness |
|---|---|
| 1. This recycling system connects the public, charities, and recycling companies to change the current opaque and unsystematic status of the recycling system and form a complete industrial chain. 2. By using China Post to promote recycling, the scope of recycling is wider, and donations are more convenient and convincing. 3. Fiber recycling factories can manufacture recycled products with higher added value in a more targeted manner, thereby increasing the public's acceptance of second-hand recycled products. 4. The information interaction platform in the charity recycling system relies on the already mature WeChat platform, which is convenient and efficient to use and has high social interaction. | 1. The scope of the small program in the charity market is limited to a certain number of people, requiring certain online operation skills, and it is more suitable for young people. 2. The profit cost is relatively high, and it needs to invest a lot of money in the initial stage for product development and the construction of transportation and recycling chains, and costs need to be invested to promote the mini program system |

**Table 4.** *Cont.*

| Opportunity | Threat |
| --- | --- |
| 1. Government policy support. The China Textile and Apparel Association issued the "Guiding Opinions on the Green Development of the Textile Industry during the "14th" Period", which also pointed out the slow construction of the waste textile recycling industry chain system. The 20th National Congress of the Communist Party of China pointed out that it is necessary to accelerate the green transformation of the development mode and promote the formation of green and low-carbon production methods and lifestyles. 2. The "Internet + recycling" model has more development prospects. In July 2021, the "14th Five-Year Plan for Circular Economy Development" proposed to improve the waste material recycling network, actively promote the "Internet + recycling" model, and realize online and offline collaboration. Second-hand clothing recycling public welfare projects carried out in the form of the Internet can promote the development of second-hand clothing recycling. 3. Improvement and support of public awareness of environmental protection. According to the survey, as many as 79% of the Chinese respondents have recognized the pollution of the modern fashion industry, and 93% of the respondents agree that they should buy or use sustainable products [69]. | 1. The construction of the second-hand trading market is immature, and the market promotion is not perfect. As a result, the Chinese people still have reservations about the acceptance of second-hand products. Therefore, although a large number of people agree with the sustainable model of recycling and reuse, they seldom buy directly for recycling. 2. The emergence of the new epidemic has made people pay more attention to the hygiene of products. The successive closures of second-hand clothing rental platforms also show that the epidemic has a greater impact on reusable products. 3. The existence of profitable recycling companies. The method of directly repaying money and profits by profitable recycling companies is also very attractive, which will have a certain impact on charity recycling. |

## 6. Results

The recycling of post-consumer textile waste is an emerging industry with abundant resources, low investment and significant benefits. It can not only solve the problem of resource shortage, but also reduce the environmental pollution of the textile industry, bringing great economic and social benefits. At present, China is taking the development of a circular economy as an important task, and actively establishing a recycling system for waste textiles is essential. Therefore, it is very necessary to study the recovery and reuse system of post-consumer textile waste.

Through the analysis of the current situation and difficulties of the recycling and reuse of post-consumer textile waste in China, combined with the excellent cases of recycling and reuse of post-consumer textile waste abroad, it is initially proposed to build a charitable market institution system under the supervision of the government. Combined with the contemporary mainstream online platform—WeChat applet—it would expand the publicity of the system among the public, improving the enthusiasm of the people to engage with this. The goal is to make the post-consumer textile waste recycling and reuse system more complete and transparent, and let the concept of sustainable development win great popular support of the public. It is believed that the recycling of post-consumer textile waste will have better progress in the future.

**Author Contributions:** Conceptualization, B.X., Q.C., B.F., R.Z. and J.F.; Formal analysis, B.X.; Writing—original draft, B.X. and Q.C.; Writing—review & editing, Q.C., B.F., R.Z. and J.F. All authors have read and agreed to the published version of the manuscript.

**Funding:** This work was funded by the Natural Science Foundation of Shanghai (21ZR1400100), Fundamental Research Funds for the Central Universities (2232020 D-45 and 2232020 E-06), by Shanghai Style Fashion Design and Value Creation Knowledge Service Center (ZX201311000031), by Arts & Humanities Research Council of UK (AH/T011483/1), by Ningbo Municipal Bureau of Foreign Experts Affairs Project (ZJFFZB-2020-102), by Ningbo Advanced Textile Technology and Apparel CAD Key Laboratory Open Fund Project (2022ZDSYS-A-003).

**Institutional Review Board Statement:** Not applicable.

**Informed Consent Statement:** Not applicable.

**Data Availability Statement:** Not applicable.

**Conflicts of Interest:** The authors declare no conflict of interest.

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
