# Peer review of "Current Situation and Construction of Recycling System in China for Post-Consumer Textile Waste"

_sustainability, doi:10.3390/su142416635_

Round 1

Reviewer 1 Report

·       Lines 28 – 34:

As 28 is shown in Figure 1, a single cotton T-shirt consumes 2,700 liters of water, and people only drink about 1,000 liters a year.

·       The comparison of water consumer in current formulation is not objective, please reason scientifically how 2.700l of water are used for a single T-shirt and how it is comparable with water consumer by people.

“Around 750 million people in the world do not have drinking water, but every ton of textiles causes 200 metric tons of water pollution. For every kilogram of cloth made, 23 kilograms of CO2 are released.”

·       Please use scientific facts and reasons instead of emotional arguments i.e., describe how does textile manufacture pollutes the water and how the reduction of the polluted water would help providing drinking water in the regions, where it is not available now.

“Total emissions exceed that of international flights and ships. In addition, about 250,000 chemicals are used in the dyeing and finishing of various textiles.”

·       Add sources of the used statistics.

·       Figure 1: the title is unintelligible

·       How much of the textile waste is being landfilled and how much incinerated? Please add corresponding statistics.

·       Generall remark about “Introduction”

o   Please work out the red thread, there is a lot of different information in the introduction part, which is not relevant for the topic of the paper e.g., why do you write about production of textiles, if the paper deals with post-consumer waste?

·       Lines 106 – 107:

For the post-consumer textile waste exported abroad, after the domestic post-con- 106 sumer textile waste recycling company collects it, it will firstly classify, clean, and sterilize 107 the waste clothing.

·       Please add statistics, how much of the waste is exported and in which countries

·       Table 2. Comparative analysis of recycling methods and related cases: Please describe the mentioned methods in more details, especially their use for textile recycling

General remark: since a proof-of-concept has not been carried out for the proposed solution, please do a SWOT analysis for the proposed approach.  

Reviewer 2 Report

Sometimes the author does not leave space after the dot in the end of the sentences, like page 2, line 56 and 59.

Page 3, line 107, reformulate the sentences without repeated "it" after the comma.

Page 3, line 108: Why the author write "textile waste will be resold and then sold ..."? What does they want to say?

Page 3, line 114: What does the authors want to say with: "Subsubsection" Lack of Recycling Channels?

Page 3 line 121: What does the authors want to say with: "some consumers will Unwanted clothes ..."?

Page 3 line 124: It's not clear the sentences: The clothes put into the old clothes recycling bin basically cannot know the whereabouts of the clothes.

Paragraph starting from line 127 -134, is not link with the previous text. Moreover in the same paragraph the author say: "so it will be difficult for people to distinguish... That is not true, its very easy as one get, and the other sell.

Figure 2 is not clear and readable.

Page 6, line 221, what does the authors want to say with "down jackets"

Table 1, why the authors mention "Yuan"

Table 2, Make the separation to make it more readable. At last coloumn do not use capital letters for "processing" and "developed by".

Line 468 and 511 show error for the references.

Line 492 reformulate the sentences by not using WeChat too many times.

Line 536 remove "In", after acknowledgments

Reviewer 3 Report

The Authors point to the textile industry as one of the most polluting sectors. They outlined the behavior of textile consumers. These people produce almost as much waste of these materials annually as they purchase new ones. Disposal of these materials is mainly through landfilling or incineration. They point out, too slow construction of the chain system of the textile waste recycling industry. Post-consumer textile waste should be utilized through the implementation of a circular economy toward waste reduction and recycling of material resources. They analyzed the behavior of the Chinese population in reusing this waste. At the same time, they compared China's recycling status with that of other countries around the world. Many countries have implemented or are preparing to implement Extended Producer Responsibility (EPR) for recycling, such as the Nordic Swan Ecolabel. Examples of national textile recycling systems include Switzerland and Belgium. There are various forms of recycling and reuse through shared consumption, leasing, renting, or donation. China has become a forerunner in online clothing rental. The most common model for waste recovery and reuse is post-consumer recycling in the form of energy recovery, mechanical, physical and chemical recycling methods. China lacks adequate regulations and standards for the recycling and reuse of used textiles. This affects the industrial export chain of used clothing to Southeast African countries. This causes sanitation problems and environmental pollution during the processes of obtaining this waste. The Authors presented tips and solutions for the use of post-consumer textile waste based on the closed waste-resource chain and based on their own research of the charitable market pattern based on government support.

Specific Comments:

Literature presented in the text inconsistent with General Considerations.

No citation in the text of the literature 62.

Fig. 4. - which means resue - here should be "reuse"

Lines 387-388 According to these classifications, there are several suggestions for the flow direction of recycled clothing: (end of sentence) - perhaps another designation?

Line 511 - Error! Reference source not found. What is the literature 67 about? - it is available online and listed in the literature index.

Round 2

Reviewer 1 Report

Dear Authors, 

thank you for the revision of the manuscript. Alls of the relevant points are addressed in the revised manuscript.